# Polynomial Alternatives to Softmax in Transformers

## Abstract

Transformers have rapidly become the backbone of modern machine learning, with attention mechanisms, most often implemented with a softmax activation, at their core. The softmax function is typically motivated by its ability to produce a row-wise probability distribution over the attention matrix, yielding sparse patterns that align with the intuition of attending to different input tokens. In this paper, we uncover an additional and previously overlooked role of softmax: it implicitly regularizes the Frobenius norm of the attention matrix, which contributes to stabilizing training. This observation prompts a fundamental question: are the inductive biases imposed by softmax: positivity, normalization, and sparsity, truly necessary for effective transformer training? To answer this, we explore alternative activations, focusing on polynomial functions that preserve the regularization effect while introducing fundamentally different inductive biases. Through theoretical analysis, we show that specific polynomial activations can serve as viable substitutes for softmax, supporting stable training and strong performance despite abandoning its conventional properties. Extensive experiments across a range of transformer architectures and applications validate our findings, providing new insights into the design of attention mechanisms.

## 1 Introduction

Transformer architectures (Vaswani et al., 2017) have become the foundation of state-of-the-art models across natural language processing (NLP) (Vaswani et al., 2017; Devlin et al., 2018; Zhuang et al., 2021; Zhen et al., 2022), computer vision (Dosovitskiy et al., 2020; Carion et al., 2020; Liu et al., 2021; Touvron et al., 2021), and robotics (Fu et al., 2024; Maiti et al., 2023; Salzmann et al., 2020). At the core of these models lies the softmax attention block, which assigns relative importance to tokens and enables transformers to capture long-range dependencies more effectively than recurrent or convolutional architectures, particularly at scale.

Softmax self-attention is appealing because it satisfies three widely discussed properties: (1) non-negativity of attention weights, (2) row-wise normalization that ensures weights sum to one (and thus admit a probabilistic interpretation), and (3) sparsity, which promotes focus on a small set of relevant tokens. These properties are often considered fundamental to effective attention (Bahdanau et al., 2014; Zhen et al., 2022), though the evidence for this view is largely empirical rather than theoretical. Despite recent explorations of alternative activations (Shen et al., 2023; Fang et al., 2022; Correia et al., 2019), softmax has remained dominant because of its strong empirical performance and interpretability.

In this paper, we revisit these assumptions and ask:

> Do attention mechanisms in transformers truly require non-negativity, normalization, and sparsity to perform well?

We present a new perspective suggesting that softmax's effectiveness arises not solely from these canonical properties, but also from its implicit regularization of the Frobenius norm of the attention matrix during training. Building on this insight, we show that simple polynomial activations, although they do not satisfy non-negativity, normalization, or sparsity, can induce a similar form of regularization and achieve competitive or even superior performance across multiple tasks. Rather

than challenging the value of softmax, our results highlight that different inductive biases can also support strong performance, expanding the design space of attention mechanisms beyond the conventional probabilistic view.

Our contributions are as follows:

1. We provide a theoretical analysis showing that softmax implicitly regularizes the Frobenius norm of the attention matrix, offering an explanation for its effectiveness that goes beyond non-negativity, normalization, and sparsity.

2. We propose polynomial activations as alternatives to softmax, demonstrating that they can induce similar regularization effects without adhering to the standard constraints, and achieve competitive performance across tasks such as image classification, object detection, instance segmentation, long range sequence modeling and language modeling.

By revisiting foundational assumptions, our work deepens the understanding of attention mechanisms and opens new directions for designing effective alternatives to softmax.

## 2 RELATED WORK

**Attention activations.** A variety of alternative activations for attention mechanisms have been explored in recent literature. Shen et al. (2023) showed that ReLU activations outperform softmax in long-sequence tasks, such as document translation. Of particular relevance to our work, Wortsman et al. (2023) demonstrated that scaling ReLU by the inverse of sequence length can surpass softmax in certain vision applications, emphasizing the importance of correct activation scaling. In this paper, we show that softmax inherently applies such a scale through its normalization, and we derive theoretical principles that motivate polynomial activations with scalings proportional to the square root of the sequence length. Other studies have proposed alternatives with varying motivations. Banerjee et al. (2020) used Taylor series approximations of softmax, achieving superior performance in image classification. Wang et al. (2021) introduced periodic activations to improve gradient flow in attention layers. Koohpayegani & Pirsiavash (2024) showed that $l^1$ normalization applied to linear attention mechanisms can yield on par performance to softmax's on three distinct vision transformers. Distinct from these works, our approach establishes a clear theoretical link between the Frobenius norm of the attention matrix and the input sequence length. Leveraging this insight, we design polynomial activations that break three canonical properties of softmax—non-negativity, row normalization, and sparsity—while still achieving competitive performance.

**Attention mechanisms.** Numerous strategies have been proposed to improve the efficiency and scalability of transformers by reducing computational overhead and rethinking attention mechanisms. The Data-Efficient Image Transformer (DeiT) (Touvron et al., 2021) leverages distillation tokens to achieve competitive performance without relying on large datasets. The Cross-Covariance Image Transformer (XCiT) (Ali et al., 2021) introduces cross-covariance attention, enabling efficient spatial interactions with reduced complexity. The Swin Transformer (Liu et al., 2021) employs a hierarchical architecture with shifted window-based self-attention to enhance scalability for vision tasks. The Nyströmformer (Xiong et al., 2021) approximates full self-attention using the Nyström method, reducing its complexity from quadratic to near-linear. Similarly, the MLP-Mixer (Tolstikhin et al., 2021) replaces self-attention entirely with multi-layer perceptrons for spatial and channel mixing. In this work, we demonstrate that polynomial-based attention activations can be incorporated into standard architectures and achieve softmax-level performance, offering competitive alternatives even without the conventional constraints of non-negativity, normalization, and sparsity.

## 3 PRELIMINARIES AND NOTATION

In this section we outline the definition of a transformer via the transformer block and set the notation of various mathematical quantities we will be using in future sections. For more details on transformers the reader can consult Vaswani et al. (2017); Dosovitskiy et al. (2020).

Transformer architectures comprise of transformer blocks, defined as follows. A transformer block is a mapping $\mathbf{T} : \mathbb{R}^{N \times D} \to \mathbb{R}^{N \times D}$ defined as

$$\mathbf{T}(x) = \mathbf{F}(\mathbf{A}(x) + x) \tag{1}$$

where $\mathbf{F}$ is a feedforward MLP with a residual connection and $\mathbf{A}$ is an attention head.

The attention head $\mathbf{A}$ is defined as follows: It comprises of three learnable matrices, a query ($q$), key ($k$) and value ($v$) defined by: $q = QX$, $k = KX$, $v = VX$ for an input sequence $X \in \mathbb{R}^{N \times D}$ with $Q, K \in \mathbb{R}^{D \times d}$ and $V \in \mathbb{R}^{D \times M}$. The attention head $\mathbf{A}(X)$ is then defined by

$$\mathbf{A}(X) = \phi(\mathcal{S}(q, k))v \tag{2}$$

where $\mathcal{S}$ is a similarity transformation and $\phi$ is an activation function. The most common used $\mathcal{S}$ is the dot-product: $\mathcal{S}(q, v) = qk^T$, known as self-attention, and will be the one we focus on in this paper. The most common activation function $\phi$ that is used by authors is softmax. This leads to the most common form of the attention head given by

$$\begin{aligned}
\mathbf{A}(X) &= \mathbf{softmax}\left(\frac{qk^T}{\sqrt{d}}\right)v \\
&= \mathbf{softmax}\left(\frac{XQK^T X^T}{\sqrt{d}}\right)XV.
\end{aligned} \tag{3}$$

The function $\mathbf{softmax}$ is the matrix softmax map that applies the usual softmax function row-wise:

$$\mathbf{softmax}\left(\begin{bmatrix} x_{11} & \cdots & x_{1n} \\ \vdots & \vdots & \vdots \\ x_{n1} & \cdots & x_{nn} \end{bmatrix}\right) = \begin{bmatrix} \frac{e^{x_{11}}}{\sum_{j=1}^n e^{x_{1j}}} & \cdots & \frac{e^{x_{1n}}}{\sum_{j=1}^n e^{x_{1j}}} \\ \vdots & \vdots & \vdots \\ \frac{e^{x_{n1}}}{\sum_{j=1}^n e^{x_{nj}}} & \cdots & \frac{e^{x_{nn}}}{\sum_{j=1}^n e^{x_{nj}}} \end{bmatrix} \tag{4}$$

The factor $\frac{1}{\sqrt{d}}$, as explained in Vaswani et al. (2017), is a scaling to prevent the gradients of softmax from being too small. For the theoretical analysis in this paper we will only use the dot-product similarity $qk^T$ and call the $N \times N$ matrix $softmax(qk^T)$ the *softmax self-attention matrix*. In the experiments, section 6, we will empirically validate our theoretical framework on more general softmax attention blocks used in state of the art transformers such as DeiT (Touvron et al., 2021), Swin Transformer (Liu et al., 2021) and XciT (Xiong et al., 2021).

For general transformer architectures, multiple heads $\mathbf{A}_i$ for $1 \leq i \leq n$ are used. Each attention head is defined by equation 3 and then all outputs of each attention head are concatenated together before going into the feedforward layer.

We will need notation for the derivative of the matrix softmax map defined by equation 4. Given a matrix $A \in \mathbb{R}^{N \times N}$ we can differentiate the matrix map $\mathbf{softmax}$ at $A$ and obtain the gradient linear map $\nabla \mathbf{softmax}(A) : \mathbb{R}^{N \times N} \to \mathbb{R}^{N \times N}$ that is defined by the formula

$$\nabla \mathbf{softmax}(A) := \mathbf{Jsoftmax}(A)^T \tag{5}$$

where $\mathbf{Jsoftmax}(A)$ is the Jacobian of $\mathbf{softmax}$ at $A$.

Given a matrix $A \in \mathbb{R}^{n \times m}$, we denote its Frobenius norm by $||A||_F$. Additionally, we use the notation $\mathbb{E}$ to represent the expectation of a random variable, where the specific random variable being considered will be clear from the context.

## 4 THEORETICAL ANALYSIS

### 4.1 IMPLICIT REGULARIZATION OF SOFTMAX

This section presents a theoretical result showing that the softmax activation imposes control over the Frobenius norm of the self-attention matrix in a way that grows sub-linearly with the input sequence's token length. Additionally, we demonstrate that the gradient of the softmax with respect to the self-attention matrix also exhibits a similar degree of regularity. While previous work has analyzed the regularity of softmax self-attention through the lens of the Lipschitz constant (Kim et al.,

2021; Castin et al., 2023), our theorem offers a novel perspective by directly linking the Frobenius norm regularity to the token length. This provides insights into how self-attention activations should scale with token length to maintain stability during training, especially with gradient descent-based algorithms.

**Theorem 4.1.** *Let* $\mathbf{softmax} : \mathbb{R}^{N \times N} \to \mathbb{R}^{N \times N}$ *be the matrix softmax map defined by equation 4 and let* $\nabla \mathbf{softmax}(A) : \mathbb{R}^{N \times N} \to \mathbb{R}^{N \times N}$ *denote the gradient of* $\mathbf{softmax}$ *at* $A \in \mathbb{R}^{N \times N}$. *We then have the following bounds on the Frobenius norms*

$$||\mathbf{softmax}(A)||_F \leq \sqrt{N} \tag{6}$$

$$||\nabla \mathbf{softmax}(A)||_F \leq 2\sqrt{N}. \tag{7}$$

The key implication of theorem 4.1 is that during the training of a transformer with softmax self-attention, the Frobenius norm of each softmax self-attention matrix remains bounded by a value that grows as $\mathcal{O}(\sqrt{N})$. This ensures that backpropagation through the weights of the self-attention matrix does not lead to excessively large gradients. The proof hinges on the fact that the row normalization inherent in softmax effectively controls the Frobenius norm. For a detailed proof see section A.1.1.

### 4.2 POLYNOMIAL ACTIVATIONS FOR ATTENTION

In section 4.1, we demonstrated that softmax implicitly regularizes the Frobenius norm of the self-attention matrix. Building on this, we now show that by scaling specific polynomial activations, a similar regularization effect on the Frobenius norm can be achieved in expectation, closely replicating the impact of softmax.

**Theorem 4.2.** *Let* $X \in \mathbb{R}^{N \times D}$ *and* $Q, K \in \mathbb{R}^{D \times d}$ *be i.i.d random variables distributed according to* $X \sim \mathcal{N}(0, \sigma_x)$ *and* $Q, K \sim \mathcal{N}(0, \sigma_t)$. *We have the following expectations of the Frobenius norms of powers of the* $N \times N$ *matrix* $(XQK^T X^T)^p$ *for* $p \geq 1$

$$\mathbb{E}\left|\left|\left(\frac{XQK^T X^T}{\sqrt{d}}\right)^p\right|\right|_F \leq \mathcal{O}(N) \tag{8}$$

By scaling such an activation by $\frac{1}{\sqrt{N}}$ we can obtain a $\mathcal{O}(\sqrt{N})$ bound.

**Corollary 4.1.** *Assume the same conditions as in theorem 4.2. Then*

$$\mathbb{E}\left|\left|\frac{1}{\sqrt{N}}\left(\frac{XQK^T X^T}{\sqrt{d}}\right)^p\right|\right|_F \leq \mathcal{O}(\sqrt{N}). \tag{9}$$

Corollary 4.1 establishes that activations of the form $\phi(x) := \frac{1}{\sqrt{N}} x^p$ provide a level of regularization, in expectation, similar to that of softmax when applied to the self-attention matrix. The proof of theorem 4.2 can be found in appendix A.1.2. The next property we want to prove is one similar to the gradient bound obtained in theorem 4.1. Since the self-attention matrix has parameters given by the queries $Q$ and keys $K$ (Vaswani et al., 2017), this implies that during the training of a transformer the $Q$ and $K$ matrices are the only aspects of the self-attention matrix that get updated. Therefore, we compute a derivative bound with respect to the $Q$ and $K$ derivatives.

**Theorem 4.3.** *Let* $X \in \mathbb{R}^{N \times D}$ *and* $Q, K \in \mathbb{R}^{D \times d}$ *be i.i.d random variables distributed according to* $X \sim \mathcal{N}(0, \sigma_x)$ *and* $Q, K \sim \mathcal{N}(0, \sigma_t)$. *Then the expectation of the of the derivative of the matrix* $\frac{(XQK^T X^T)^p}{\sqrt{d}}$ *w.r.t the* $Q$ *parameter matrix for* $p \geq 1$ *is given by*

$$\mathbb{E}\left|\left|\frac{\partial}{\partial Q}\left(\frac{(XQK^T X^T)^p}{\sqrt{d}}\right)\right|\right| \leq \mathcal{O}(N) \tag{10}$$

**Corollary 4.2.** *Assume the same condition as in theorem 4.3. Then*

$$\mathbb{E}\left|\left|\frac{1}{\sqrt{N}}\frac{\partial}{\partial Q}\left(\frac{(XQK^T X^T)^p}{\sqrt{d}}\right)\right|\right| \leq \mathcal{O}(\sqrt{N}). \tag{11}$$

An analogous estimate holds for derivatives with respect to the $K$ matrix. The proof of theorem 4.3 can be found in appendix A.1.2.

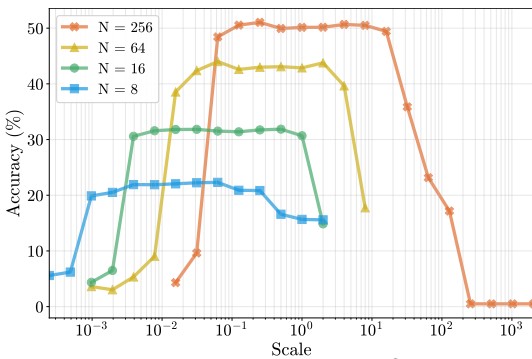

Figure 1: Training ViT-Tiny with the activation $\phi(x) = x^3$ with different sequence lengths and different scales. As the sequence length gets larger, the $k$ scale (x-axis) needed to obtain good accuracy when using $\frac{1}{k}x^3$ as an activation increases validating the theory from section 4.2.

*Remark* 4.1. corollary 4.1 and corollary 4.2 suggest that polynomial activations of the form $\phi(x) = \frac{1}{\sqrt{N}}x^p$, with $p > 0$, can achieve performance comparable to softmax when applied to self-attention matrices. We point out that both corollaries rested on the assumption that $X$, $Q$ and $K$ are i.i.d random variables. In general, this is only true at initialization when training a transformer, see Albert et al. (2025). Although this is a limitation in the theory the experiments in section 6 show that this insight can be used effectively to develop new attention blocks that perform comparable to softmax yet violate the three conditions of positivity, normalized rows summing to 1 and sparsity showing that attention blocks do not need to be modeled as a probability distribution.

## 5 TESTING THE THEORY

In this section we test the theory developed in section 4.2 on small vision transformers. We will consider the activation $\phi(x) = \frac{1}{k}x^3$, where $k > 0$ is a fixed scale, as this activation clearly violates the three key conditions of softmax based attention; positivity, normalization and sparsity. We found that polynomials $\frac{1}{k}x^p$ for $p > 3$ did not perform well during training as they witnessed a gradient vanishing problem due to the fact that the function $x^p$ for $p$ large have very small values around 0.

The first experiment we conducted was to test how the Top-1% accuracy changes for a ViT-Tiny vision transformer (Steiner et al., 2021), trained on the Tiny-Imagenet dataset (Stanford University CS231n Course, 2015), as we change the sequence length $N$ of the input and the scale predicted in corollaries 4.1 and 4.2 when using the activation $\frac{1}{k}x^3$. The standard ViT-Tiny model comprises 12 transformer layers, each equipped with 3 attention heads, with each head having dimension 64. We considered four different input sequence lengths $N$ of sizes 256, 64, 16 and 8. For each such sequence length, we ran a ViT-Tiny architecture with the activation $\frac{1}{k}x^3$ where $k$ ranged from roughly $10^{-3}$ to $10^3$. According to the theory developed in section 4, the Frobenius norm of $\frac{1}{\sqrt{N}}x^3$ scales according to $\mathcal{O}(\sqrt{N})$. Thus the best accuracy should occur when $k = \mathcal{O}(\sqrt{N})$ and should degrade for other values due to training instability. fig. 1 shows the results of the experiment, we note that the $x-axis$ plots the values of $k$. We see that as the sequence length increases the factor of $k$ needs to increase so that the activation $\frac{1}{k}x^3$ performs well on the ViT-Tiny architecture as predicted by the theory developed in corollaries corollary 4.1 and 4.2.

In a second experiment we decided to compare the performance of the original ViT-Tiny architecture, that uses an input sequence length of 256, with a softmax activation and a polynomial activation on the Tiny-ImageNet dataset. In corollary 4.1 and corollary 4.2 it was pointed out that the scaling of the polynomial is important to keep the Frobenius norm of the polynomial attention matrix from becoming too large. The results of those corollaries suggested that a scale to use is $\frac{1}{\sqrt{N}}$, $N$ being the sequence length, which in this case is $\frac{1}{\sqrt{256}} = \frac{1}{16}$. We therefore decided to to compare the three activations softmax, $x^3$ and $\frac{1}{16}x^3$. Each was trained for 200 epochs using the AdamW optimizer. To begin with we computed the Frobenius norm of the attention matrix throughout training averaged over all the heads in layers 2, 7 and 12 of the ViT-Tiny architecture when trained on the Tiny-ImageNet dataset. fig. 2 plots the results. We see from the figure that the Frobenius norm of of

the $x^3$ archicture is much larger than softmax but scaling it by $\frac{1}{16}$ brings it down to softmax levels. Similarly, fig. 3 shows the Jacobian's Frobenius norm, where scaling also brings the norms closer to softmax, ensuring more stable gradients. Further plots for other layers are in section A.2.3. table 1 presents the final Top-1% accuracy achieved by each activation function. Notably, $\frac{x^3}{16}$ delivers the best performance. In contrast, the unscaled $x^3$ activation yields significantly lower Top-1% accuracy, underscoring the importance of incorporating an appropriate scaling factor.

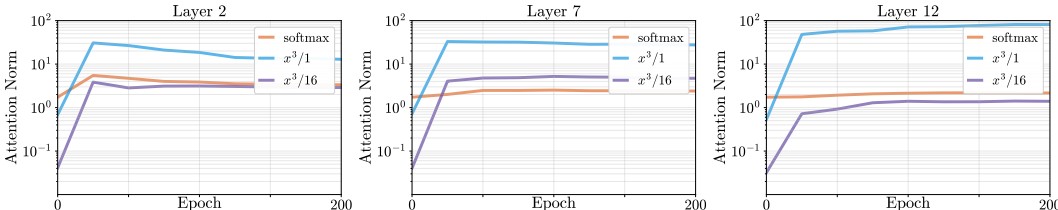

Figure 2: Frobenius norm of the self-attention matrix with three different activations in layer 2, 7 and 12 of the ViT-Tiny architecture during training.

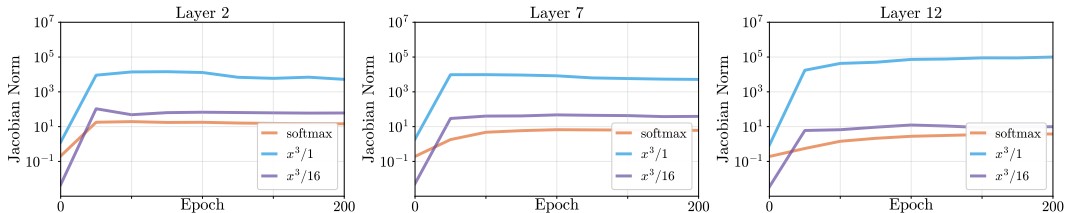

Figure 3: Frobenius norm of the jacobian of the self-attention matrix with three different activations in layer 2, 7 and 12 of the ViT-Tiny architecture during training.

Table 1: Comparison of Top-1% accuracy on Tiny-ImageNet between softmax and polynomial activations.

|  | softmax | $\frac{x^3}{16}$ | $x^3$ |
|---|---|---|---|
| Top-1% accuracy | 50.26 | **50.5** | 45.3 |

## 6 EXPERIMENTS

In this section, we evaluate a simple polynomial activation as an alternative to the standard softmax across a range of transformer applications commonly studied in the literature. The goal is to empirically challenge the conventional softmax properties, positivity, row-normalization, and sparsity—and examine whether these conditions are truly necessary for good transformer performance.

Building on the theoretical foundations from section 4, we focus on the cubic polynomial $x^3$ as a test case. Notably, $x^3$ introduces both positive and negative values, does not normalize rows to sum to 1, and generally produces dense attention matrices—violating all three traditional softmax conditions.

We consider two scaling strategies for $x^3$:

1. **Fixed scale:** Following section 4, we scale $x^3$ by the inverse square root of the sequence length (which is fixed throughout training), as theory suggests this maintains optimization stability.

2. **Learned scale:** Recognizing that the assumption of i.i.d. normal-distributed $Q$, $K$, $V$, used in section 4, holds primarily at initialization, we also explore a learnable scale. This learnable scale is initialized as above but optimized during training.

Our experiments are not designed to achieve state-of-the-art results. Instead, they aim to question the softmax paradigm and show that alternative activations, even those violating softmax's traditional properties, can still lead to effective transformer models.

## 6.1 IMAGE CLASSIFICATION

We conducted an image classification task using various vision transformer architectures from the literature on the ImageNet-1k dataset. For this task, the standard sequence length employed by the vision transformers on the ImageNet-1k dataset is 196.

We trained all models on the ImageNet-1k dataset from scratch and report Top-1 accuracy on the validation set. We examined our approach along with the following four transformer architectures to show its generalization, ViT-B (Dosovitskiy et al., 2020), DeiT (Touvron et al., 2021), Swin Transformer (Liu et al., 2021), XCiT (Ali et al., 2021). Each transformer was trained following the approach in each paper. The results are shown in table 2. As can be seen from the table the $x^3$ activation with a learned scale performed the best and the one with a fixed scale performed comparable to softmax. On the other hand $x^3$ (with no scale) underperformed on all ViTs showing how important the theory on scaling as developed in section 4 is. For a discussion on other polynomials and linear attention see section A.2.4.

Table 2: Comparisons of pretraining models with different activation functions on ImageNet-1k. We report top-1 classification accuracy (%).

| | Models | | | |
| --- | --- | --- | --- | --- |
| | ViT-B | DeiT-B | Swin-B | XciT-M |
| softmax | 80.3 | 81.5 | 83.5 | 82.7 |
| $x^3$ + fixed scale | 80.2 | 81.4 | 83.6 | 82.8 |
| $x^3$ + learned scale | 80.3 | 81.6 | 83.6 | 82.9 |
| $x^3$ | 78.1 | 78.5 | 79.9 | 79.5 |

**Visualizing attention heads:** Softmax attention traditionally satisfies three key properties: positivity, row normalization, and sparsity. In contrast, the polynomial activation $x^3$ (with or without positive scaling) takes positive values for $x > 0$ and negative values for $x < 0$, thus violating these softmax constraints. To better understand how this affects attention patterns, we analyzed the self-attention matrices of ViT-B models trained with the $x^3$ + learned scale activation and compared them to those using softmax. We visualized heatmaps of the attention matrices after convergence, focusing on two representative layers and heads, and averaged over a fixed batch of 128 samples. fig. 4 shows results for layer 2, head 8, where the $x^3$ + learned scale activation produces attention scores with both positive and negative values, unlike softmax. Similarly, fig. 5 illustrates distinct patterns for layer 12, head 6, highlighting how the two activations differ in learned attention distributions.

**Interpretability.** Even with these differences, the $x^3$ + learned scale activation still learns meaningful attention patterns. For instance, in fig. 4 (left), we observe 14 bands about the diagonal, indicating that the head has learned to attend to patches in the same image row as the query patch, sufficient for effective image classification. This suggests that the sign of attention values (positive or negative) is not inherently critical for attention allocation. Further, fig. 5 (left) reveals vertical lines, showing attention that depends solely on key positions, independent of the query position. Smaller key indices receive higher attention weights, focusing model capacity where it matters most for classification tasks. These findings demonstrate that, despite deviating from softmax properties, the $x^3$ + learned scale activation enables the model to discover effective attention patterns. We noticed similar attention patterns for the $x^3$ + fixed scale activation.

## 6.2 OBJECT DETECTION AND INSTANCE SEGMENTATION

In this section, we evaluate transfer learning by fine-tuning an ImageNet-pretrained XCiT-S12 on COCO 2017 (Lin et al., 2014) for object detection and segmentation. We integrate XCiT as the

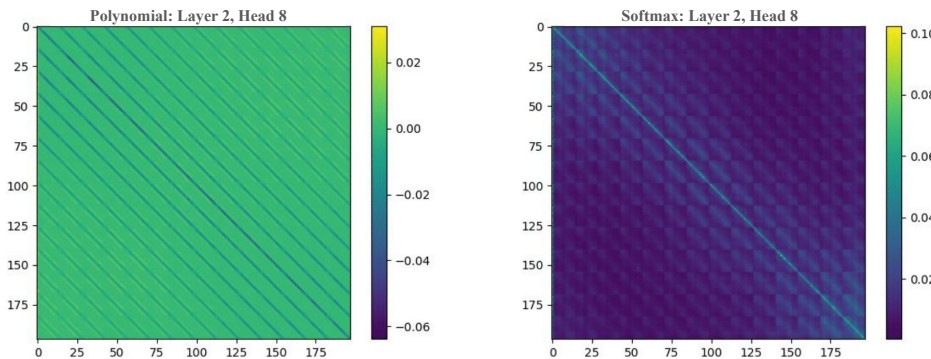

Figure 4: Heat maps of the self-attention matrix in layer 2, head 8, of a ViT base architecture, comparing $x^3$ + learned (left) and softmax (right) activations after training. The stark difference in self-attention patterns between the two activations is evident, showing distinct distributions across input tokens.

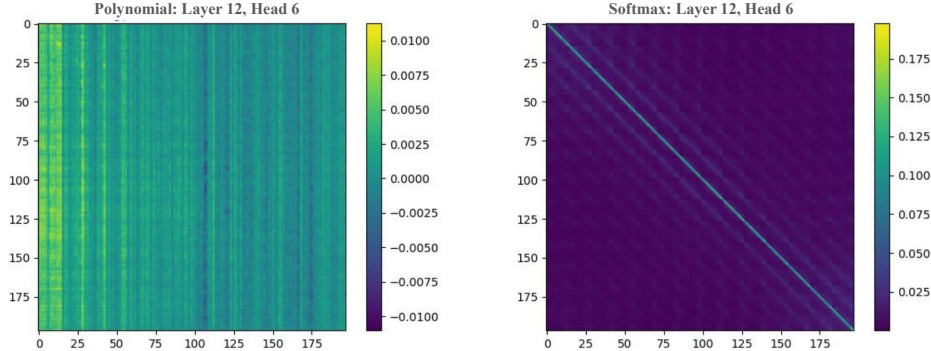

Figure 5: Heat maps of the self-attention matrix in layer 12, head 6, of a ViT base architecture, comparing $x^3$ + learned scale (left) and softmax (right) activations after training. The contrast in self-attention patterns between the two activations is clearly visible.

backbone in Mask R-CNN (He et al., 2017) with an FPN, adapting its columnar design by extracting multi-scale features. Models were trained with softmax, $x^3$ + fixed scale, and $x^3$ + learned scale (initialized at $1/14$). Without scaling, $x^3$ failed to converge, consistent with our theory in section 4.2. Results in table 3 show that $x^3$ + learned scale outperforms softmax, while $x^3$ + fixed scale remains comparable.

Table 3: COCO object detection and instance segmentation performance on the mini-val set. All backbones are pretrained on ImageNet-1k, and use Mask R-CNN model. $AP^b$: Average Precision for bounding box predictions, $AP^b_{50/75}$: Average Precision at an IoU threshold of 0.50/0.75 for bounding box predictions, $AP^m$: Average Precision for mask predictions, $AP^m_{50/75}$: Average Precision at an IoU threshold of 0.50/0.75 for mask predictions.

|  | $AP^b$ | $AP^b_{50}$ | $AP^b_{75}$ | $AP^m$ | $AP^m_{50}$ | $AP^m_{75}$ |
|---|---|---|---|---|---|---|
| softmax | 44.9 | 66.1 | 48.9 | 40.1 | 63.1 | 42.8 |
| $x^3$ + fixed scale | 44.8 | 66.3 | 49.1 | 40.2 | 63.1 | 43.0 |
| $x^3$ + learned scale | 45.1 | 66.5 | 49.4 | 40.4 | 63.2 | 43.1 |

## 6.3 Nyströmformer on LRA benchmark

Modeling long sequences is essential for transformers, as tasks such as text understanding require capturing dependencies across hundreds or thousands of tokens. The Nyströmformer (Xiong et al., 2021) addresses this by approximating self-attention with the Nyström method, achieving near-linear complexity while retaining long-range modeling power. To evaluate our approach, we trained models on five datasets from the Long Range Arena (LRA) suite (Tay et al., 2020): ListOps, Text Classification, Retrieval, Image Classification, and Pathfinder.

On each dataset we compared softmax, $x^3$, and $x^3$ with a learned scale. For the scaled variant, initialization followed the sequence length of the original Nyströmformer (i.e., $1/\sqrt{L}$ for sequence length $L$). We adopted the training protocol of Xiong et al. (2021). Results are reported in table 4, showing that $x^3$ + learned scale consistently outperforms softmax, $x^3$ + fixed scale performs comparably, and unscaled $x^3$ underperforms across all tasks.

Table 4: Comparisons of Nyströmformer models with different activation functions on the LRA benchmark. We report the accuracy (%).

|  | ListOps | Text | Retrieval | Image | Pathfinder |
|---|---|---|---|---|---|
| softmax | 37.1 | 63.8 | 79.8 | 39.9 | 72.9 |
| $x^3$ + fixed scale | 37.2 | 63.7 | 80.0 | 39.9 | 72.9 |
| $x^3$ + learned scale | 37.5 | 63.9 | 81.1 | 40.1 | 73.1 |
| $x^3$ | 32.3 | 62.0 | 78.5 | 38.1 | 67.9 |

## 6.4 Language Modeling

In section A.2.5, we evaluate polynomial activations in the language modeling setting and show that, when appropriately scaled, a cubic polynomial can achieve performance comparable to softmax.

## 7 Limitations

While our work introduces polynomial activations as alternatives to softmax, limitations remain. First, our theoretical framework is developed specifically for dot-product self-attention and may not directly extend to other attention variants, such as additive attention or kernelized approximations. Exploring these extensions could yield further insights. Second, although our experiments cover multiple architectures and tasks, they are restricted to models of up to 100 million parameters due to resource constraints. Assessing the scalability of polynomial activations in large-scale transformers with billions of parameters is an important direction for future work.

## 8 Conclusion

This work questioned whether transformer activations for attention must produce sparse probability distributions. We introduced a theoretical framework analyzing the Frobenius norm of the self-attention matrix, which suggests key scaling properties for activations in attention mechanisms. We proved that specific polynomial activations, which behave very differently from softmax, satisfy these properties. Through extensive experiments across a variety of transformer applications, we demonstrated that these alternative activations not only compete with but can outperform softmax, offering a new perspective on attention mechanisms in transformers. [1]

---

[1]Digital writing assistance tools were used for grammar and formatting. No large language models were involved in the research itself, and all scientific contributions are original work by the authors.

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

# A  APPENDIX

## ETHICS STATEMENT

All experiments in this study were conducted on publicly available benchmark datasets. No human subjects, personal information, or sensitive data were used. The methods introduced are intended purely for fundamental research in machine learning.

## REPRODUCIBILITY STATEMENT

We have taken care to ensure the reproducibility of all results presented in this paper. Where external code was used, explicit references are provided, and all experimental settings, including hardware details, are documented in the appendix. In addition, full proofs of all theoretical results are included in the appendix to enable independent verification.

## USE OF LLMS

We used digital writing assistance tools for grammar and formatting. No large language models were involved in conducting the research, and all scientific contributions are the original work of the authors.

### A.1  THEORETICAL ANALYSIS

#### A.1.1  PROOFS FOR THEOREMS IN SECTION 4.1

In this section we give the proof of theorem 4.1.

*Proof of theorem 4.1.* We will start by proving the first inequality in theorem 4.1. Given a matrix $A = (a_{ij}) \in \mathbb{R}^{N \times N}$ we have that

$$
\mathbf{softmax}\left(\begin{bmatrix} a_{11} & \cdots & a_{1n} \\ \vdots & \vdots & \vdots \\ a_{n1} & \cdots & a_{nn} \end{bmatrix}\right) = \begin{bmatrix} \frac{e^{a_{11}}}{\sum_{j=1}^n e^{a_{1j}}} & \cdots & \frac{e^{a_{1n}}}{\sum_{j=1}^n e^{a_{1j}}} \\ \vdots & \vdots & \vdots \\ \frac{e^{a_{n1}}}{\sum_{j=1}^n e^{a_{nj}}} & \cdots & \frac{e^{a_{nn}}}{\sum_{j=1}^n e^{a_{nj}}}. \end{bmatrix} \tag{12}
$$

By definition of the Frobenius norm we then see that

$$
||\mathbf{softmax}(A)||_F^2 = \left(\frac{1}{\sum_{j=1}^n e^{a_{1j}}}\right)^2 (e^{2a_{11}} + \cdots e^{2a_{11}}) + \cdots \tag{13}
$$

$$
+ \left(\frac{1}{\sum_{j=1}^n e^{a_{Nj}}}\right)^2 (e^{2a_{N1}} + \cdots e^{2a_{NN}}) \tag{14}
$$

$$
\leq \left[\left(\frac{1}{\sum_{j=1}^n e^{a_{1j}}}\right)(e^{a_{11}} + \cdots e^{a_{11}})\right]^2 + \cdots \tag{15}
$$

$$
+ \left[\left(\frac{1}{\sum_{j=1}^n e^{a_{Nj}}}\right)(e^{a_{N1}} + \cdots e^{a_{NN}})\right]^2 \tag{16}
$$

$$
= 1 + \cdots + 1 \tag{17}
$$

$$
= N \tag{18}
$$

where the second inequality uses the fact that for non-negative numbers $a$ and $b$ we always have that $a^2 + b^2 \leq (a + b)^2$.

It then immediately follows that $||\mathbf{softmax}(A)||_F \leq \sqrt{N}$ and this proves the first inequality in the statement of theorem 4.1.

We move on to prove the second inequality in the statement of theorem 4.1. For this, let us write each entry of the matrix on the right of equation 12 as follows:

$$F_{kl} = \frac{e^{a_{kl}}}{\sum_{j=1}^{N} e^{a_{kj}}}. \tag{19}$$

By applying the chain rule we then have the following derivative formulas

$$\frac{\partial}{\partial x_{ij}} F_{ij} = F_{ij} - F_{ij}^2 \tag{20}$$

$$\frac{\partial}{\partial x_{ik}} F_{ij} = -F_{ij} F_{ik} \text{ for any } k \neq j \tag{21}$$

$$\frac{\partial}{\partial x_{kl}} F_{ij} = 0 \text{ for any } k \neq i \text{ and } l \neq j. \tag{22}$$

We can then express the gradient as

$$\nabla \mathbf{softmax}(A) = \begin{bmatrix} \nabla F_{11} \\ \vdots \\ \nabla F_{1N} \\ \nabla F_{21} \\ \vdots \\ \vdots \\ \nabla F_{NN} \end{bmatrix} \tag{23}$$

where

$$\nabla F_{ij} = \begin{bmatrix} -F_{ij} F_{i1}^2 & -F_{ij} F_{i2} & \cdots & F_{ij} - F_{ij}^2 & \cdots & F_{ij} F_{iN}. \end{bmatrix} \tag{24}$$

From these computations we see that

$$||\nabla \mathbf{softmax}(A)||_F^2 = ||\nabla F_{11}||_F^2 + \cdots + ||\nabla F_{NN}||_F^2. \tag{25}$$

We will proceed by bounding each collection $||\nabla F_{i1}||_F^2 + \cdots + ||\nabla F_{1N}||_F^2$ separately then add up all the bounds. We have

$$||\nabla F_{i1}||_F^2 + \cdots + ||\nabla F_{1N}||_F^2 = |F_{i1} - F_{i1}^2|^2 + |F_{i1} F_{i2}|^2 + \cdots + |F_{i1} F_{iN}|^2 \tag{26}$$

$$+ |F_{i2} F_{i1}|^2 + |F_{i2} - F_{i2}^2|^2 + \cdots + |F_{i2} F_{iN}|^2 \tag{27}$$

$$+ \cdots\cdots\cdots\cdots + \tag{28}$$

$$+ |F_{iN} F_{i1}|^2 + |F_{iN} F_{i2}|^2 + \cdots + |F_{iN} - F_{iN}^2|^2 \tag{29}$$

$$\leq (F_{i1})^2 (|1 - F_{i1}| + |F_{i2}| + \cdots + |F_{iN}|)^2 \tag{30}$$

$$(F_{i2})^2 (|F_{i1}| + |1 - F_{i2}| + \cdots + |F_{iN}|)^2 \tag{31}$$

$$+ \cdots\cdots\cdots\cdots + \tag{32}$$

$$+ (F_{iN})^2 (|F_{i1}| + |F_{i2}| + \cdots + |1 - F_{iN}|)^2. \tag{33}$$

We then observe that since $F_{i1} + \cdots + F_{iN} = 1$ we have that $1 - F_{ij} = 2(F_{i1} + \cdots + \widehat{F_{ij}} + \cdots + F_{iN})$ where $\widehat{F_{ij}}$ means we don't include $F_{ij}$ in the sum. This means we get the bound

$$||\nabla F_{i1}||_F^2 + \cdots + ||\nabla F_{1N}||_F^2 \leq 4F_{i1}^2 (\widehat{F_{i1}} + F_{i2} + \cdots + F_{iN}) \tag{34}$$

$$+ \cdots\cdots\cdots\cdots + \tag{35}$$

$$+ 4F_{iN}^2 (F_{i1} + F_{i2} + \cdots + \widehat{F_{iN}}) \tag{36}$$

$$\leq 4(F_{i1}^2 + \cdots F_{iN}^2) \tag{37}$$

$$= 4. \tag{38}$$

Putting all the bounds together for each of the terms $N$ terms $||\nabla F_{i1}||_F^2 + \cdots + ||\nabla F_{1N}||_F^2$ we get

$$||\nabla \mathbf{softmax}(A)||_F^2 \leq 4N \tag{39}$$

and this implies

$$||\nabla \mathbf{softmax}(A)||_F \leq 2\sqrt{N}. \tag{40}$$

This finishes the proof of theorem 4.1.

$\square$

### A.1.2 PROOFS FOR THEOREMS SECTION 4.2

In this section we will give the proof of theorems 4.2 and 4.3.

*Proof of theorem 4.2.* We will split the matrix product $XQK^TX^T$ and think of it as the product of two matrices. Suppose $\mathbf{A} \in \mathbb{R}^{N \times D} \sim \mathcal{N}(0, \sigma_1^2)$, $\mathbf{B} \in \mathbb{R}^{D \times N} \sim \mathcal{N}(0, \sigma_2^2)$ and $\mathbf{C} = \mathbf{AB}$. Each element in the matrix $\mathbf{C}$ can be written as a product of a row of $\mathbf{A}$ with a column of $\mathbf{B}$. Since expectation is linear, we need to compute the expectation of each of these elements. We do the case of the entry $c_{11}$ which is the entry in $\mathbf{C}$ in the first row and first column. For the $p = 1$ case we can then compute

$$
\begin{aligned}
\mathbb{E}(c_{11}^2) &= \mathbb{E}((\sum_{i=1}^{D} a_{1i}b_{i1})^2) \\
&= \mathbb{E}(\sum_{i=1}^{D} a_{1i}^2 b_{i1}^2 + \sum_{i=1}^{D}\sum_{j=1,j\neq i}^{D} a_{1i}b_{i1}a_{1j}b_{j1}) \\
&= \sum_{i=1}^{D} \mathbb{E}(a_{1i}^2)\mathbb{E}(b_{i1}^2) + \sum_{i=1}^{D}\sum_{j=1,j\neq i}^{D} \mathbb{E}(a_{1i})\mathbb{E}(b_{i1})\mathbb{E}(a_{1j})\mathbb{E}(b_{j1}) \\
&= D\sigma_1^2\sigma_2^2 + 0.
\end{aligned}
\tag{41}
$$

The Frobenius norm of the matrix $\mathbf{C}$ is just the sum of these values for all $N^2$ elements and this proves the $p = 1$ case.

For the case that $p > 1$ we proceed in a similar way. The key observation is that odd powers, in the matrix expansion, will have expectaion 0, so we need only consider the even powers. Therefore, suppose $\mathbf{C} = (\mathbf{AB})^p$. We will compute the expectation of the first entry $c_11 \in \mathbf{C}$:

$$
\begin{aligned}
\mathbb{E}(c_{11}^2) &= \mathbb{E}((\sum_{i=1}^{D} a_{1i}b_{i1})^{2p}) \\
&= \mathbb{E}(\sum_{i=1}^{D} a_{1i}^{2p}b_{i1}^{2p} + \sum_{i=1}^{D}\sum_{j=1,j\neq i}^{D} a_{1i}^{2p-2}b_{i1}^{2p-2}a_{1j}^2 b_{j1}^2 + \cdots).
\end{aligned}
\tag{42}
$$

Note that the first term only has a count of $D$ and the second term has a count of $D(D-1)$. Thus, we only need to consider the $\mathcal{O}(D^p)$ term where all the components have a power of 2. The count is similar to choosing $p$ items from $D$,

$$
\begin{aligned}
\mathbb{E}(c_{11}^2) &\approx \mathbb{E}(\sum_{\{i_1,\ldots,i_p\}\in\{1,\ldots,D\}} \prod_{k=1}^{p} a_{1,i_k}^2 b_{i_k,1}^2) \\
&= \binom{D}{p} \frac{2p!}{2^p} \sigma_1^{2p}\sigma_2^{2p} \\
&= \frac{D!}{(D-p)!} \frac{2p!}{p!2^p} \sigma_1^{2p}\sigma_2^{2p} \\
&= \frac{D!}{(D-p)!} \frac{2p!}{2p!!} \sigma_1^{2p}\sigma_2^{2p} \\
&= \frac{D!}{(D-p)!} (2p-1)!!\sigma_1^{2p}\sigma_2^{2p}.
\end{aligned}
\tag{43}
$$

$\frac{D!}{(D-p)!}$ can always be bounded above by $D^p$, so the expectation can be upper bounded by $D^p(2p-1)!!\sigma_1^{2p}\sigma_2^{2p}$ and thus we get a quantity of the form $\mathcal{O}(N)$.

$\square$

*Proof of theorem 4.3.* We will do the $p = 1$ case first. We proceed similar to the proof of Theorem 4.2.

$$\mathbb{E}(\|\frac{\partial XQK^TX^T}{\partial Q}\|_F^2) = \sum_{i=1}^{N}\sum_{j=1}^{N}\mathbb{E}(|\frac{\partial x_i^TQK^Tx_j}{\partial Q}|_F^2) \tag{44}$$

$$= \sum_{i=1}^{N}\sum_{j=1}^{N}\mathbb{E}(\|x_i x_j^T K\|_F^2) \tag{45}$$

$$= \sum_{i=1}^{N}\sum_{j=1}^{N}\mathbb{E}(\sum_{k=1}^{D}\sum_{l=1}^{d}(x_{ik}\sum_{m=1}^{D}x_{jm}k_{ml})^2) \tag{46}$$

$$= \sum_{i=1}^{N}\sum_{j=1}^{N}\mathbb{E}(\sum_{k=1}^{D}\sum_{l=1}^{d}x_{ik}^2(\sum_{m=1}^{D}x_{jm}k_{ml})^2) \tag{47}$$

$$= \sum_{i=1}^{N}\sum_{j=1}^{N}\mathbb{E}(\sum_{k=1}^{D}\sum_{l=1}^{d}x_{ik}^2(\sum_{m=1}^{D}x_{jm}^2k_{ml}^2 + \sum_{m=1}^{D}\sum_{n=1,n\neq m}^{D}x_{jm}k_{ml}x_{jn}k_{nl})) \tag{48}$$

$$= \sum_{i=1}^{N}\sum_{j=1}^{N}\sum_{k=1}^{D}\sum_{l=1}^{d}(\sum_{m=1}^{D}\mathbb{E}(x_{ik}^2x_{jm}^2k_{ml}^2) \tag{49}$$

$$+ \sum_{m=1}^{D}\sum_{n=1,n\neq m}^{D}\mathbb{E}(x_{ik}^2x_{jm}k_{ml}x_{jn}k_{nl})) \tag{50}$$

$$= \sum_{i=1}^{N}\sum_{j=1}^{N}\sum_{k=1}^{D}\sum_{l=1}^{d}(\sum_{m=1}^{D}\mathbb{E}(x_{ik}^2x_{jm}^2k_{ml}^2) + 0) \tag{51}$$

$$= \sum_{i=1}^{N}\sum_{j=1}^{N}\sum_{k=1}^{D}\sum_{l=1}^{d}\sum_{m=1}^{D}\mathbb{E}(x_{ik}^2x_{jm}^2k_{ml}^2) \tag{52}$$

$$= \sum_{i=1}^{N}\sum_{k=1}^{D}\sum_{l=1}^{d}\mathbb{E}(x_{ik}^2x_{ik}^2k_{kl}^2) \tag{53}$$

$$+ \sum_{i=1}^{N}\sum_{j=1,j\neq i}^{N}\sum_{k=1}^{D}\sum_{l=1}^{d}\sum_{m=1,m\neq k}^{D}\mathbb{E}(x_{ik}^2x_{jm}^2k_{ml}^2) \tag{54}$$

$$= NDd3\sigma_x^4\sigma_w^2 + N(N-1)D(D-1)d\sigma_x^4\sigma_w^2 \tag{55}$$

$$\approx N^2D^2d\sigma_x^4\sigma_w^2. \tag{56}$$

When $p > 1$ we can proceed in a similar way.

$$\mathbb{E}(\|\frac{\partial (XQK^TX^T)^p}{\partial Q}\|_F^2) = \sum_{i=1}^{N}\sum_{j=1}^{N}\mathbb{E}(\|\frac{(\partial x_i^TQK^Tx_j)^p}{\partial Q}\|_F^2) \tag{57}$$

$$= \sum_{i=1}^{N}\sum_{j=1}^{N}\mathbb{E}(\|p(x_i^TQK^Tx_j)^{p-1}\frac{\partial x_i^TQK^Tx_j}{\partial Q}\|_F^2) \tag{58}$$

$$= \sum_{i=1}^{N}\sum_{j=1}^{N}\mathbb{E}(\|p(x_i^TQK^Tx_j)^{p-1}x_i x_j^T K\|_F^2) \tag{59}$$

$$= \sum_{i=1}^{N}\sum_{j=1}^{N}\mathbb{E}(p^2(x_i^TQK^Tx_j)^{2p-2}\sum_{k=1}^{D}\sum_{l=1}^{d}(x_{ik}\sum_{m=1}^{D}x_{jm}k_{ml})^2). \tag{60}$$

We know that

$$(x_i^T Q K^T x_j)^{2p-2} = (\sum_{l=1}^{d}((\sum_{k=1}^{D} x_{ik}q_{kl}) \cdot (\sum_{m=1}^{D} x_{jm}k_{ml})))^{2p-2} \tag{61}$$

$$= (\sum_{l=1}^{d}\sum_{k=1}^{D}\sum_{m=1}^{D} x_{ik}q_{kl}x_{jm}k_{ml})^{2p-2} \tag{62}$$

$$= (\sum_{k=1}^{D}\sum_{m=1}^{D} x_{ik}x_{jm}\sum_{l=1}^{d} q_{kl}k_{ml})^{2p-2} \tag{63}$$

$$= (\sum_{k=1}^{D}\sum_{m=1}^{D} x_{ik}x_{jm}a_{km})^{2p-2}, \tag{64}$$

where $a_{km} = \sum_{l=1}^{d} q_{kl}k_{ml}$. Let $z_{ij} = \sum_{k=1}^{D}\sum_{m=1}^{D} x_{ik}x_{jm}a_{km}$ Thus we have

$$\mathbb{E}(\|\frac{\partial(XQK^TX^T)^p}{\partial Q}\|_F^2) = p^2 \sum_{i=1}^{N}\sum_{j=1}^{N}\mathbb{E}(z_{ij}^{2p-2}\sum_{k=1}^{D}\sum_{l=1}^{d}(x_{ik}\sum_{m=1}^{D} x_{jm}k_{ml})^2) \tag{65}$$

$$= \sum_{i=1}^{N}\sum_{j=1}^{N}\sum_{k=1}^{D}\sum_{l=1}^{d}(\sum_{m=1}^{D}\mathbb{E}(z_{ij}^{2p-2}x_{ik}^2x_{jm}^2k_{ml}^2) \tag{66}$$

$$+ \sum_{m=1}^{D}\sum_{n=1,n\neq m}^{D}\mathbb{E}(z_{ij}^{2p-2}x_{ik}^2x_{jm}k_{ml}x_{jn}k_{nl})) \tag{67}$$

$$= \sum_{i=1}^{N}\sum_{j=1}^{N}\sum_{k=1}^{D}\sum_{l=1}^{d}(\sum_{m=1}^{D}\mathbb{E}(z_{ij}^{2p-2}x_{ik}^2x_{jm}^2k_{ml}^2) \tag{68}$$

$$+ \sum_{m=1}^{D}\sum_{n=1,n\neq m}^{D}\mathbb{E}(z_{ij}^{2p-3}x_{ik}^2x_{jm}^2k_{ml}^2x_{jn}^2k_{nl}^2)) \tag{69}$$

$$\approx N^2 Dd(D(D^{2p-2}d^{p-1}(2p-3)!!\sigma_x^{4p}\sigma_w^{4p-2}) + 0 \tag{70}$$

$$= N^2 D^{2p}d^p(2p-3)!!\sigma_x^{4p}\sigma_w^{4p-2} \tag{71}$$

showing that we can bound the gradient by a quantity of the form $\mathcal{O}(N)$ and the proof is complete.

$\square$

## A.2 EXPERIMENTS

### A.2.1 HARDWARE

The vision transformer experiments in section 6.1, the object detection and instance segmentation experiments in section 6.2 and the Nyströmformer experiments from section 6.3 were all carried out on Nvidia A100 GPUs.

### A.2.2 EXPERIMENTAL HYPERPARAMETERS

**Vision transformers in section 6.1.** In section 6.1 we tested four different vision transformers, ViT-B (Dosovitskiy et al., 2020), DeiT-B (Touvron et al., 2021), Swin-B (Liu et al., 2021) and XCiT-M (Ali et al., 2021), with the activations softmax, $x^3$ + fixed scaling, $x^3$ + learned scaling and $x^3$. The training strategy follow the exact strategy used in each of the original papers, we used the Timm libraries to train our models (Wightman, 2019).

### A.2.3 FROBENIUS NORM COMPUTATIONS

In section 5 we showed plots of the Frobenius norm of the self-attention matrix and for the Jacobian of the self-attention matrix for softmax, $\frac{1}{14}x^3$, and $x^3$. This was done for a ViT-Tiny architecture on

the Tiny-ImageNet dataset. fig. 6 shows the plots of the Frobenius norm of the self-attention matrix for the ViT-Tiny architecture, during training, for all layers averaged over the heads within each layer. fig. 7 shows the Frobenius norm of the Jacobian of the self-attention matrix during training for each layer, averaged over the total number of heads within each layer.

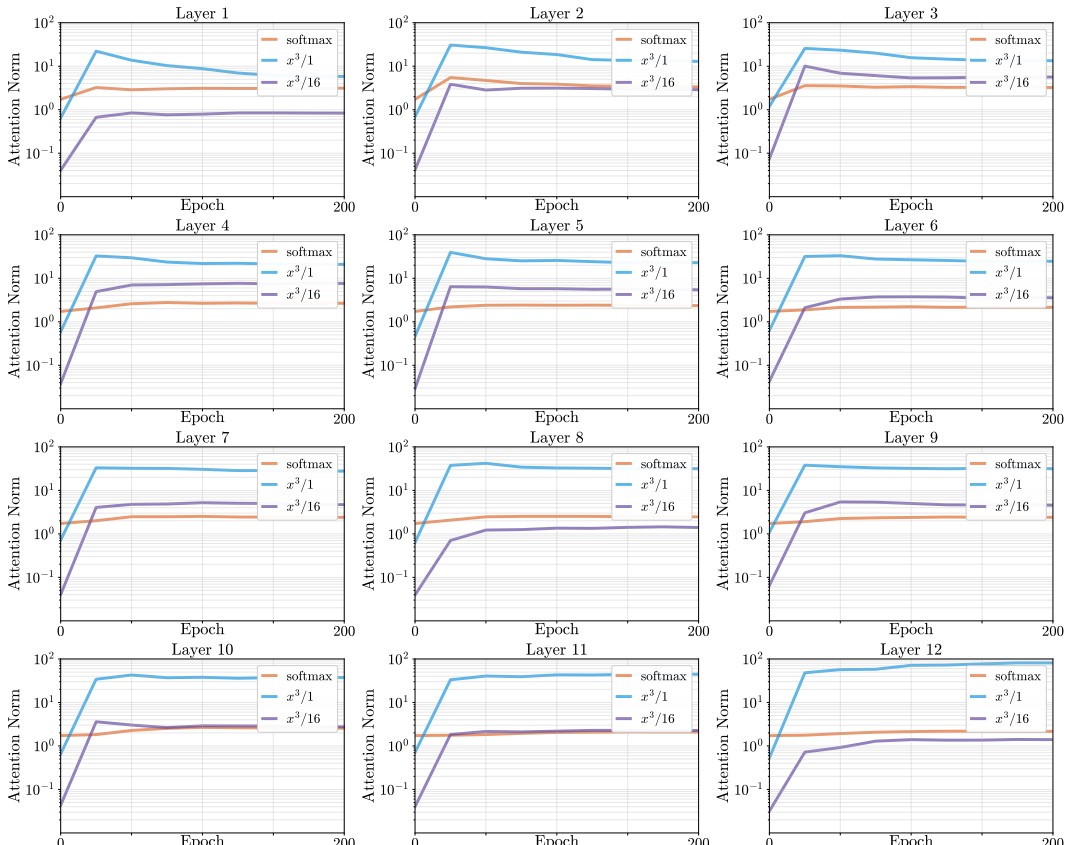

Figure 6: Frobenius norm of self-attention matrix for softmax, $\frac{1}{16}x^3$ and $x^3$ on ViT-Tiny during training on the Tiny-ImageNet dataset (zoom in for better viewing).

### A.2.4 DISCUSSION ON LINEAR ACTIVATED ATTENTION AND OTHER POLYNOMIALS ON VITS

In the experiments section 6 we showed our insights using the polynomial $x^3$ as this was a non-linear polynomial that did not satisfy the key properties that softmax did. In this section we will compare softmax with other polynomials that we also found to perform well.

The results for ViTs is shown in table 5. As we can see from that table each polynomial does far better when scaled using the theory from section 4. We note that when we tried to train polynomial higher than order 6 they did not train well. On inspecting the weights we found that several were very close to zero leading to a gradient vanishing problem. We hypothesize that this is because the transformers were randomly initialized with weights about zero. Thus taking such a weight and applying a polynomial of the form $x^k$ with $k \geq 6$ would make those weights orders of magnitude smaller, making it difficult to train after some point.

### A.2.5 LANGUAGE MODELING

We compare softmax with cubic activations under different scaling strategies to test whether the core softmax properties are necessary for strong performance on language modeling tasks. We pretrained a GPT-2 on WikiText-103 then evaluated on PTB, and PG-19, the naïve $x^3$ activation performs poorly, but both fixed- and learned-scaled variants achieve perplexities close to or better than softmax as shown table 6, table 7 and table 8. This suggests that appropriate scaling, rather

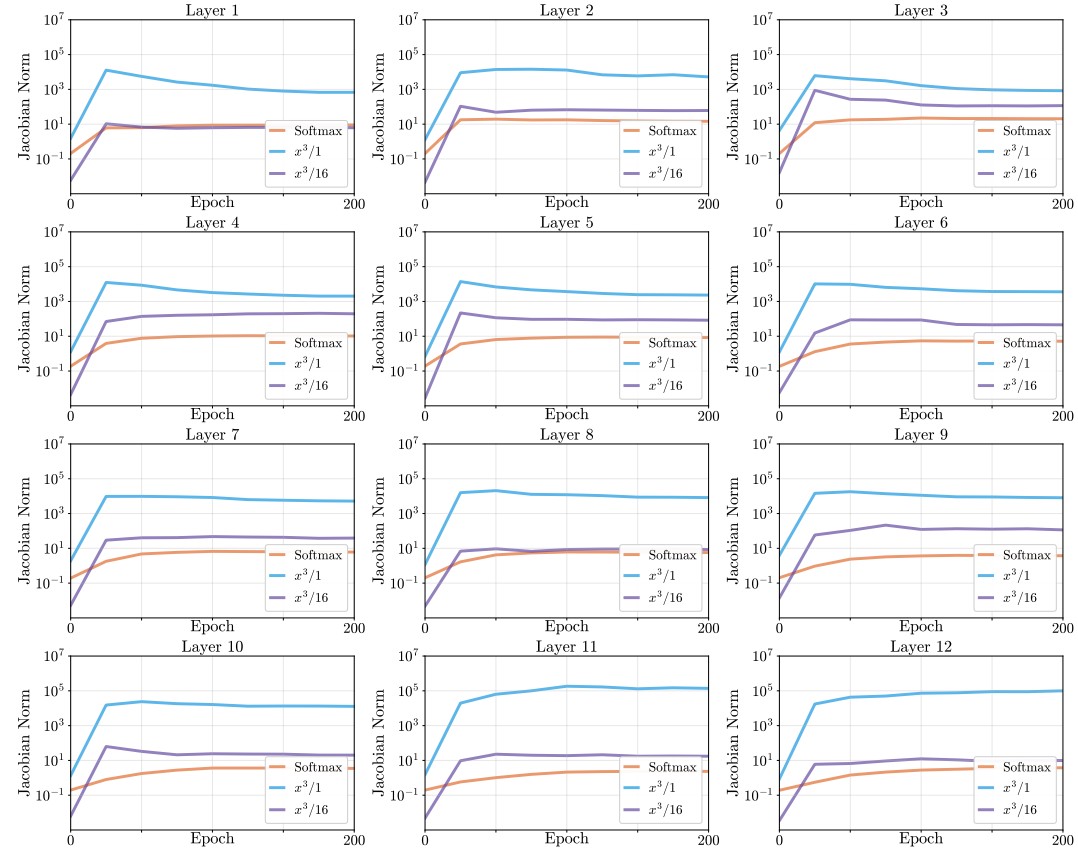

Figure 7: Frobenius norm of the Jacobian of the self-attention matrix for softmax, $\frac{1}{16}x^3$ and $x^3$ on ViT-Tiny during training on the Tiny-ImageNet dataset (zoom in for better viewing).

than strict adherence to softmax's probabilistic form, is sufficient to retain competitive performance in language modeling.

Table 5: Comparisons of pretraining models with different activation functions on ImageNet-1k. We report top-1 classification accuracy (%).

| | Models | | | |
| --- | --- | --- | --- | --- |
| | ViT-B | DeiT-B | Swin-B | XciT-M |
| softmax | 80.3 | 81.5 | 83.5 | 82.7 |
| $x$ + fixed scale | 78.4 | 79.4 | 80.6 | 80.2 |
| $x$ + learned scale | 78.7 | 79.5 | 80.7 | 80.4 |
| $x$ | 74.1 | 77.9 | 78.3 | 78.1 |
| $x^2$ + fixed scale | 80.1 | 81.5 | 83.4 | 82.5 |
| $x^2$ + learned scale | 80.3 | 81.6 | 83.5 | 82.7 |
| $x^2$ | 77.8 | 78.2 | 79.8 | 79.4 |
| $x^4$ + fixed scale | 80.3 | 81.5 | 83.7 | 82.7 |
| $x^4$ + learned scale | 80.3 | 81.6 | 83.7 | 82.8 |
| $x^4$ | 77.9 | 78.6 | 79.9 | 79.6 |
| $x^5$ + fixed scale | 80.3 | 81.4 | 83.4 | 82.5 |
| $x^5$ + learned scale | 80.3 | 81.5 | 83.4 | 82.6 |
| $x^5$ | 77.7 | 78.0 | 79.4 | 79.5 |

Table 6: Perplexity of GPT-2 models pretrained on WikiText-103.

| Model | Perplexity |
| --- | --- |
| softmax | 45.2 |
| $x^3$ | 49.8 |
| $x^3$ + fixed scale | 45.4 |
| $x^3$ + learned scale | 45.0 |

Table 7: Perplexity of GPT-2 models evaluated on PTB.

| Model | Perplexity |
| --- | --- |
| softmax | 50.4 |
| $x^3$ | 55.7 |
| $x^3$ + fixed scale | 50.3 |
| $x^3$ + learned scale | 50.1 |

Table 8: Perplexity of GPT-2 models evaluated on PG-19.

| Model | Perplexity |
| --- | --- |
| softmax | 55.1 |
| $x^3$ | 59.8 |
| $x^3$ + fixed scale | 55.1 |
| $x^3$ + learned scale | 54.9 |

