# OpenReview forum: "Polynomial Alternatives to Softmax in Transformers"
_ICLR.cc/2026/Conference — ICLR 2026 Conference Withdrawn Submission_

### Official Review · Reviewer_1GoG · 2025-10-24

**Soundness:** 2
**Presentation:** 3
**Contribution:** 2
**Rating:** 2
**Confidence:** 4

**Summary:**

This paper challenges the assumption that *the inductive biases imposed by softmax: positivity, normalization, and sparsity are necessary for transformer training*. They instead argue that the bounded norm of the attention matrix are indeed important. Building upon this observation, they propose using scaled polynomial activations that preserve the regularization effect on the attention matrix. Empirically, they observe that when using the proper scaling in the polynomial activations, the norm of the attention matrix and its Jacobian are bounded, and the training performance can match softmax transformer on small scale datasets.

**Strengths:**

**Challenge a core assumption in transformer design** This paper empirically shows that when using a polynomial activation function, the performance of the transformer can match the softmax transformer in certain cases. This shows sign of potential to replace the softmax module whose complexity is $N^2$ during inference, and have a more efficient transformer.

**Simple yet effective replacement** The proposed polynomial activations are extremely simple, parameter-free, and analytically tractable. It's surprising that such simple replacement would work.

**Extensive empirical validations** This paper presents empirical validation of their hypothesis across different domains: image classification, object detection, instance segmentation, and text understanding.

**Weaknesses:**

**Weak theoretical contribution** The bound on softmax and its Jacobian (see Theorem 4.1) holds for any inputs and weight matrices. However, the ones for polynomial activation (see Theorem 4.2 and 4.3) only hold in expectation when Q,K,V matrices follow Gaussian distribution. Thus, there is no guarantee that such properties will hold once the training starts. Though the authors conduct experiments on ViT-Tiny to verify this property empirically, I find the experimental setup too limited because the experiments are done using small models with selected layers, it is unclear whether similar properties hold for large-scale models.

**Missing discussions on gradient vanishing** If I understand correctly, the authors provide upper bounds on the Jacobian of softmax and polynomial activations to show that one can prevent the gradient explosion. However, it is unclear whether the softmax and polynomial activations can cope with *gradient vanishing*. I recommend the authors should at least mention it in the paper to make the story complete.

**Missing intuition of using polynomial activations** Can the authors provide intuitions of using polynomial activations instead to check the conditions? Moreover, the polynomial activations propose a dense attention matrix, and the interpretability and representational implications are underexplored.

**Questions:**

**Question 1** In the experimental setup for Figure 3, I'm wondering if the authors have tried using different initialization scale relevant to the length of the sequences for polynomial activations to see if the performance of unscaled polynomial activations networks can match the performance of softmax.

**Question 2** Why do the authors choose layer 2,7,12 in Figure 2 and Figure 3? Is there any specific reason to choose these layers? The indices look random to me.

**Question 3** Why not using weight decay on the Q, K matrices? If one aims to bound the attention matrix and its Jacobian, the most naive and straightforward way to deal with the boundedness of the Q, K matrices is to add weight decay on them. Moreover, the authors are using AdamW whose default hyperparameters set weight decay to be 0.01 in pytorch. In the experiment, do the authors employ this hyperparameter? If the answer is yes, then the empirical observation in Figure 3 should also be acknowledged to the weight decay.

---

### Official Review · Reviewer_H56s · 2025-10-29

**Soundness:** 2
**Presentation:** 2
**Contribution:** 1
**Rating:** 2
**Confidence:** 4

**Summary:**

This paper questions the necessity of three canonical properties (nonnegativity, normalization, and sparsity) for effective transformer attention, which are traditionally enforced by the softmax activation. The authors show that softmax also implicitly regularizes the Frobenius norm of the attention matrix and its gradient. This insight motivates an investigation of polynomial alternatives that intentionally relax these properties. The paper presents theoretical results demonstrating that, under certain conditions on the input distribution, polynomial activations with appropriate scaling can retain similar norm bounds and gradient stability to softmax. Extensive empirical evaluations across image classification, object detection, segmentation, and language modeling demonstrate that these alternatives can achieve performance comparable to, or even surpass, softmax-based transformers.

**Strengths:**

- The theoretical insight that softmax regularizes the Frobenius norm of attention matrices and their gradients offers a novel perspective for understanding the attention mechanism.

- The results are presented across diverse and well-established datasets and tasks.

- The theorems are stated clearly, proved in detail, and meaningfully relate matrix norms and activation scaling to stability and training dynamics.

**Weaknesses:**

- The core theoretical results on polynomial activations (Theorems 4.2 and 4.3) rely on the assumption of i.i.d. Gaussian inputs and weights which is not maintained during training, as briefly acknowledged in Remark 4.1. This weakens the universality of the proposed scaling prescriptions.

- The theoretical results indicate that the softmax function guarantees boundedness of the matrix softmax map regardless of the input and weight distributions, while the polynomial alternative requires i.i.d. Gaussian inputs and weights. This suggests that softmax is inherently more robust than polynomial activations in maintaining a bounded Frobenius norm. The experiments further support this, showing that all results are highly sensitive to the choice of the scaling factor $k$.

- By relaxing nonnegativity and normalization constraints, the interpretability of attention weights as probability distributions is lost. Moreover, the polynomial activation does not provide any alternative meaningful interpretation beyond maintaining Frobenius norm boundedness.

- Using polynomial activations may amplify the effects of very large or very small entries, potentially leading to gradient vanishing or explosion issues.

- Scalability is not demonstrated at truly large scales, where the experiments are limited to models of up to approximately 100 million parameters. It remains unclear whether polynomial activations exhibit stable behavior in larger transformer models or real-world deployments, where emergent behaviors may differ.

- In several experiments, the improvements achieved by the scaled polynomial activations are marginal or even absent.

- Minor errors: Line 114 presents the incorrect order of matrix multiplications for $q$, $k$, and $v$, and does not clearly explain the meaning of each dimension.

**Questions:**

- What are the potential deviations in real-world scenarios for input and weight distributions, and how do these deviations affect the stability of the matrix softmax map?

- How does the proposed method scale to larger transformer models?

- The attention heatmaps (Figures 4 and 5) are interesting; however, the paper only briefly comments on their qualitative differences compared to softmax. A deeper qualitative or quantitative interpretability analysis would provide richer insights into the trade-offs of non-probabilistic attention mechanisms.

---

### Official Review · Reviewer_AF8b · 2025-10-30

**Soundness:** 1
**Presentation:** 2
**Contribution:** 1
**Rating:** 2
**Confidence:** 4

**Summary:**

This paper asks a foundational question: why is the softmax so effective in attention mechanisms? The authors argue that its success arises not from commonly cited inductive biases (sparsity, non-negativity, and normalization), but from an implicit regularization of the Frobenius norm of the attention matrix. They theoretically show that this norm remains bounded under softmax, and that polynomial activations can achieve a similar property under i.i.d. assumptions. They propose such polynomial activations as viable alternatives and evaluate them across several transformer architectures and tasks, including vision, object detection, and language modeling.

**Strengths:**

1. The paper is well-written and easy to read.
2. It raises an important and underexplored question about the true source of softmax's effectiveness.
3. It provides a simple, easily implementable alternative (polynomial activations) and evaluates it across multiple architectures and modalities.
4. The experiments are consistent with the proposed scaling law, showing internal coherence between the theory and empirical findings.

**Weaknesses:**

The central weakness is that the paper’s theoretical contribution is superficial and speculative. The claimed link between the boundedness of the Frobenius norm and the empirical performance of attention mechanisms is neither rigorously established nor conceptually convincing. The analysis consists of elementary norm inequalities that do not yield any substantive understanding of optimization stability, generalization, or inductive bias. The argument that this “implicit regularization” explains the success of softmax is therefore unsubstantiated. Remark 4.1, in particular, makes an unjustified logical leap: the boundedness results from corollaries 4.1 and 4.2 do not in any way imply comparable performance between polynomial activations and softmax. Moreover, the empirical finding that higher-order polynomials fail due to vanishing gradients directly contradicts the claimed mechanism, revealing that norm control alone is not a meaningful predictor of training behavior. As a result, the theory is descriptive at best and fails to provide any explanatory or predictive value. To make the work credible, the authors would need to ground the theoretical claims in actual optimization dynamics, provide causal evidence that Frobenius-norm control stabilizes or improves learning, and clearly define what “performance” means beyond final accuracy metrics. In its current form, the work does not meet the standard of theoretical or empirical rigor expected for ICLR.

**Questions:**

1. Can you provide a more rigorous justification for why boundedness of the Frobenius norm should lead to improved training stability or generalization?

2. How does the proposed regularization relate to other forms of stabilization in transformers (e.g., normalization layers, gradient clipping)?

---

### Official Review · Reviewer_LrVA · 2025-11-01

**Soundness:** 2
**Presentation:** 3
**Contribution:** 2
**Rating:** 2
**Confidence:** 3

**Summary:**

The paper proposes replacing softmax in transformer attention with polynomial activations (primarily $x^2$ and $x^3$) motivated by a new analysis: both softmax and (properly scaled) polynomial score maps yield Frobenius-norm bounds of $O(N)$ for the attention matrix and its Jacobian, which the authors argue explains training stability. With this lens, they claim softmax’s usual inductive biases (positivity, normalization, sparsity) are not necessary. Experiments on ImageNet classification, COCO detection/segmentation, and LRA show performance roughly on par with softmax; quadratic with learned/fixed scaling tends to match or slightly exceed softmax, while unscaled cubic can be unstable.

**Strengths:**

- The paper presents a clear theoretical statement connecting attention score mappings to Frobenius-norm regularization, with analogous gradient bounds.

- The work broadens the design space for attention activations by demonstrating that strict probabilistic constraints are not required for stable training.

- Empirically well-organized comparisons across several model families (ViT/DeiT/Swin/XCiT; Nyströmformer; Mask R-CNN backbones).

**Weaknesses:**

- My main concern is that the motivation from theoretical analysis is not compelling for the replacement of softmax activation by the polynomial ones. Showing similar $O(N)$ Frobenius bounds establishes viability, not advantage. The analysis does not explain why polynomial activations should improve optimization, generalization, calibration, or length generalization relative to softmax.

- The cubic activation is unstable. In particular, $x^3$ is highly sensitive to score magnitude (vanishing for small scores, exploding for large), and the paper itself reports non-convergence without careful scaling. This weakens the claim that “polynomials” are robust substitutes; effectively only the quadratic looks serviceable.

- Empirical gains are marginal and within variance. Across ImageNet/COCO/LRA, the quadratic with proper scaling is typically at parity or shows very small improvements over softmax. There is no multi-seed variance analysis, so many differences may be statistically insignificant.

- No comparisons to other non-softmax activations. There is also no study of length generalization or robustness.

- Practical concerns left open.How masking, causal decoding, and pointer-style tasks behave without normalization/positivity is not evaluated.

**Questions:**

- How do polynomial activations compare against other alternatives of softmax under identical training?

- How robust are results to model scale and much longer sequences?

- For tasks where normalization/positivity matter (e.g., pointer networks, alignment tasks, seq2seq decoding), do polynomials degrade performance or require extra fixes?

- Do polynomial activations change throughput/memory enough to justify adoption even at parity?

---

### Note · Authors · 2025-11-24

I have read and agree with the venue's withdrawal policy on behalf of myself and my co-authors.